# Temporal demands of elite fencing

**Rafael Tarragó[1], Lindsay Bottoms[2]\*, Xavier Iglesias[1]**

**1** Grup de Recerca en Ciències de l'Esport INEFC Barcelona (GRCEIB), Institut Nacional d'Educació Física de Catalunya (INEFC), Universitat de Barcelona, Barcelona, Spain, **2** Centre for Research in Psychology and Sport Sciences, University of Hertfordshire, Hatfield, United Kingdom

\* l.bottoms@herts.ac.uk, lindsaybottoms@gmail.com

## Abstract

There is no published study that analyses the temporal demands of fencing, in its six Olympic modalities in the same high-level competition. The only existing publications date back decades, with already obsolete regulations, or focus on a specific modality. The aim of this study is to carry out a chronometric analysis of elite fencing at the 2014 Fencing World Championships to determine the existence of differences between its weapons (épée, foil and sabre) and between genders. A total of 96 elite fencers (48 women) of 24 different nationalities were analysed in the final rounds (direct elimination). We undertook observational methodology based on an open and systematic process with ideographic, point and multi-dimensional design. We analysed the work time, rest time, total time, number of stops and work-to-rest ratio. The Lince software was used to record and analyse video data. The coding and recording process of the 83 fights analysed generated a matrix of 5900 records. The quality of the data was verified by checking the validity of the observation instrument and the intraobserver reliability. The mean work time (between Allez and Halte) was 17.9 ± 3.1 s for épée, 5.8 ± 2.5 s for foil and 1 .7 ± 0.4 s for sabre. The fight work-to-rest ratios recorded were 1:0.9, 1:2.6 and 1:9.2 for épée, foil and sabre respectively. The results showed significant differences (p < .05) in the work and rest times between the three fencing weapons. However, there were no differences between fencers of the same weapon according to gender.

## Introduction

In the literature, the analysis of work and rest times of fencing in different modalities presents some confusion. There is no study that has carried out an exhaustive and comparative analysis of the six Olympic modalities in the same high-level competition. There are various records published from over 30 years ago [1–4], with more from 2 decades ago [5, 6] and finally some more recently published [7–9]. Throughout the history of fencing, its regulations have often been modified. The most significant change for sabre competitions was after the appearance of electrical equipment in 1988 and the banning of crossing the legs during attacks in the 1990s.

The literature regarding the temporal dynamics of fencers is scarce. The first contributions were made from the 1969 German Championships [4], the 1981 World Championships [2, 3], and in Canadian competitions [1]. In the 1990s, Roi and Pittaluga [6] carried out a temporary

**Data Availability Statement:** All relevant data are within the paper.

**Funding:** This study was funded by the Ministerio de Universidades (Spain) in the form of a Mobility Stays for Professors & Researchers in Foreign Centers grant to XI [PRX21/00210] and by the

Ministerio de Cultura y Deporte, Consejo Superior de Deportes (Spain) and European Union in the form of a grant to XI & RT [EXP_74847; 2023].

**Competing interests:** The authors have declared that no competing interests exist.

analysis of women's épée, and Iglesias and Rodríguez [10] carried it out for men's épée and women's foil. The first study that performs a temporal analysis of women's sabre was provided by Aquili et al. [7] and of the first for women's foil was Wylde et al. [9]. The influence of the time factor on the effectiveness of fencing actions has also been analysed in previous studies [8, 10].

Timing is a determining aspect of performance in fencing, therefore, in this article we will analyze many of the timing elements that influence the dynamics of a fight between fencers, such as: the total duration of the bouts, the relationship between the work and non-activity times in the rounds (work-to-rest ratio), the breaks in the three periods of 3 minutes in the rounds of direct elimination, among others.

Turner et al. [11–13] describes the importance of the time factor in the performance of fencers and suggest specific physical training based on the work-rest relationship linked to the literature published to date. The authors propose future work should observe the work/rest ratio of fencing in an important competition, with data separated by weapon and gender, therefore this is the purpose of the current study. The aim of this study is, therefore, to carry out a chronometric analysis of elite fencing in a world championship to determine the existence of differences between its weapons and between the gender modalities. Results from this study will help inform coaches and aid the creation of training plans that can better prepare their fencers for the demands of the bouts in elite level competition.

## Materials and methods

### Participants

Ninety-six fencers of 24 nationalities were observed, 16 of each fencing modality. We registered 83 bouts (last 32, last 16, quarter finals, semi finals and final) of the 2014 World Fencing Championship. The videos belonged to the International Fencing Federation (https://www.youtube.com/@FIEvideo). The study was conducted according to the Declaration of Helsinki, and approved by the Catalan sports research ethics committee (0099S/2912/2010 2607/LA).

### Instruments

The observation instrument used was ESGRIMOBS [14] and it was registered with LINCE (v.1.1) [15]. Data was analysed using Microsoft® Excel® 2016 and SPSS Statistics (v.22, Armonk, NY: IBM Corp.).

### Procedure

The unit of observation was the time between the calls of *Allez* and *Halte* [16]. Bouts were analysed in slow motion, when necessary, instead of in real time.

We analysed:

○ Total Bout Time (TBT): the length of time from the start to finish of the bout

○ Work Time (WT): the total of the intervals of time between the *Allez* and *Halte*

○ Total Pause Time (TPT): the sum of seconds that the chronometer is stopped during the bout (PRT+PTBP)

○ Period Rest Time (PRT): the time that elapses between each *Halte* and *Allez* of the same period

○ Pause Time Between Periods (PTBP): the rest time that passes between the end of a period and the beginning of the next one

○ Halte (n): Number of arrests of the bout (Halte) during the active phase of the assault (including the *Halte* at the end of each period)

○ Average *Allez* Time by bout (AAT): the average time from the referee giving the signal of *Allez* until *Halte*

○ Average *Halte* Time by bout (AHT): the average time that elapses each time the chronometer is stopped, without taking account the PTBP

○ Work-to-rest ratio (W/R): the comparison between the time that the chronometer is running that is stopped during the bout. Specifically, the calculation is made by dividing the AHT by the global AAT of each weapon or period and the resulting value becomes the REST ratio equivalent to each WORK unit.

○ Periods: 1st Period (1PER), 2nd Period (2PER), 3rd Period (3PER), Tie Extra Period (ET_A), Non-Combat Extra Period (ET_B).

## Statistical analysis

Descriptive statistics are presented as means and standard deviations. The Shapiro-Wilks test determined the non-normal distribution. The test of differences for gender, weapon and periods were analysed through non-parametric Kruskal-Wallis test with pairwise comparison by Mann-Whitney U test, using the Bonferroni correction.

## Results

A summary of the data can be seen in Table 1. We observe how the Effective Combat Time (WT·[WT+PRT]$^{-1}$), in the overall number of rounds is 32.7±21.7%. In women's épée (62.5 ±4.5%) and men's épée (53.6±9.7%), the percentage of working time is higher than that for women's foil (31.8±8.2%) and men's (25.5±9.3%), and both épée and foil show higher activity records than women's sabre (11.5±1.7%) and men's sabre (9.4±3.2%). In the WT comparison, the 6 fencing modalities present significant differences when compared to each other, except between genders of the same weapon (p < .05). In the PRT comparison, the 6 fencing modalities present significant differences when compared to each other, except between genders of the same weapon and WF vs ME, MF vs MS, MF vs WS, WF vs MS and WF vs WS (p < .05).

**Table 1. Temporal structure of the final phase of 2014 World Fencing Championships.**

|  | Women's épée | | | Men's épée | | | Women's foil | | | Men's foil | | | Women's sabre | | | Men's sabre | | | Total | | |
|---|---|---|---|---|---|---|---|---|---|---|---|---|---|---|---|---|---|---|---|---|---|
| Bouts (n) | 15 | | | 15 | | | 11 | | | 12 | | | 15 | | | 15 | | | 83 | | |
| TBT (s) | 841.0 | ± | 214.3 | 1016.1 | ± | 284.6 | 992.3 | ± | 316.9 | 977.4 | ± | 322.0 | 715.7 | ± | 156.9 | 832.8 | ± | 286.0 | 888.3 | ± | 279.3 |
| WT (s) | 433.8 | ± | 103.8 | 444.3 | ± | 88.0 | 280.6 | ± | 112.5 | 233.9 | ± | 123.8 | 70.3 | ± | 16.6 | 63.3 | ± | 14.6 | 253.8 | ± | 179.9 |
| TPT (s) | 407.2 | ± | 121.2 | 571.9 | ± | 215.9 | 711.6 | ± | 253.0 | 743.5 | ± | 236.9 | 645.4 | ± | 143.2 | 769.5 | ± | 276.8 | 634.5 | ± | 241.8 |
| PRT (s) | 262.6 | ± | 76.5 | 417.2 | ± | 190.0 | 615.7 | ± | 236.9 | 677.7 | ± | 210.9 | 548.6 | ± | 139.3 | 673.4 | ± | 279.4 | 523.3 | ± | 244.9 |
| PTBP (s) | 144.6 | ± | 51.5 | 154.7 | ± | 39.1 | 95.9 | ± | 43.2 | 65.8 | ± | 65.4 | 96.8 | ± | 15.6 | 96.2 | ± | 39.6 | 111.2 | ± | 52.8 |
| *Halte* (n) | 24.2 | ± | 4.3 | 24.9 | ± | 5.3 | 43.5 | ± | 7.0 | 44.8 | ± | 5.5 | 37.4 | ± | 7.7 | 42.3 | ± | 15.1 | 35.5 | ± | 11.9 |
| AAT (s) | 17.8 | ± | 2.8 | 18.1 | ± | 3.5 | 6.5 | ± | 2.6 | 5.1 | ± | 2.3 | 1.9 | ± | 0.3 | 1.6 | ± | 0.5 | 8.7 | ± | 7.5 |
| AHT (s) | 12.4 | ± | 2.0 | 18.6 | ± | 6.2 | 14.5 | ± | 3.6 | 15.6 | ± | 3.8 | 15.5 | ± | 2.5 | 16.6 | ± | 3.4 | 15.6 | ± | 4.2 |
| W/R | 1: 0.7 | | | 1: 1.0 | | | 1: 2.2 | | | 1: 3.0 | | | 1: 8.2 | | | 1: 10.4 | | | 1: 1.8 | | |

Note. Values are mean ± SD. Total Bout Time (TBT), Work Time (WT), Total Pause Time (TPT), Period Rest Time (PRT), Pause Time Between Periods (PTBP), Average Allez Time by bout (AAT), Average Halte Time by bout (AHT) and Work-to-rest ratio (W/R).

If we analyze the total relationship between work time (WT) and total breaks (TTP), we observe how the relationship with respect to the total time of the fight (WT/TBT) is lower than the previous values, both in the total of the modalities (28.1±18.0%) and in each one of them (WE 52.0±4.9%; ME 45.2±7.5%; WF 28.7±7.3%; MF 23.7±7.8%; WS 9.9±1.3%; MS 8.1±2.5%). In the comparison of TBT in the 6 modalities, no significant differences were observed except between WS vs ME, WE, MF and WF (p < .05).

In a global analysis, by weapon, we observe that the AAT values are 17.9 ± 3.1s for épée and 5.8 ± 2.5s for foil and 1.7 ± 0.4s for the sabre. On the other hand, the AHT records are 15.5 ±5.5s; 15.1 ±3.7s; 16.0 ±3.0s respectively. Finally, the W/R in épée is 1:0.9, in foil 1:2.6 and in sabre 1:9.2.

We can see evidence of significant differences in the chronometric analysis of fencing bouts depending on the weapon in the different temporal variables: TBT (sabre < épée & foil), WT (sabre < foil < épée), TPT (épée < foil & sabre), PRT (épée < foil & sabre), PTBP (sabre & foil < épée), *Halte* (épée < sabre < foil). The most interesting records in this temporal are the significant differences that are evident in the AAT, with much higher records being observed in men's (18.2±3.4s) and women's (17.8±2.8s) épée compared to those observed in men's (5.1 ±2.3s) and women's (6.5±2.6s) foil. In these two modalities, there are no differences between genders in the same weapon. Sabre clearly shows lower AAT values than foil and épée (AAT: sabre < foil < épée) (p < .05). On the whole, women have lower records in Total Pause Time (TPT), Period Rest Time (PRT) and Average Halte Time by bout (AHT) compared to men (p < .05).

Fig 1 shows us the relationship between work time and break time in fights, in this case considering both recovery times during fight periods, as well as rest times between statutory periods. The inverse relationship between % Work Time (WT·[WT+TPT]$^{-1}$) and % Total Rest Time (TPT·[WT+TPT]$^{-1}$) in matches of the final phase of the 2014 World Fencing Championship can be observed in Fig 1.

In women's épée, values close to a 50% relationship between work time (WT) and rest time (TPT) are observed in the three regular periods. During the one-minute extra time working time values are slightly lower. From the comparison between the three periods of 3 minutes,

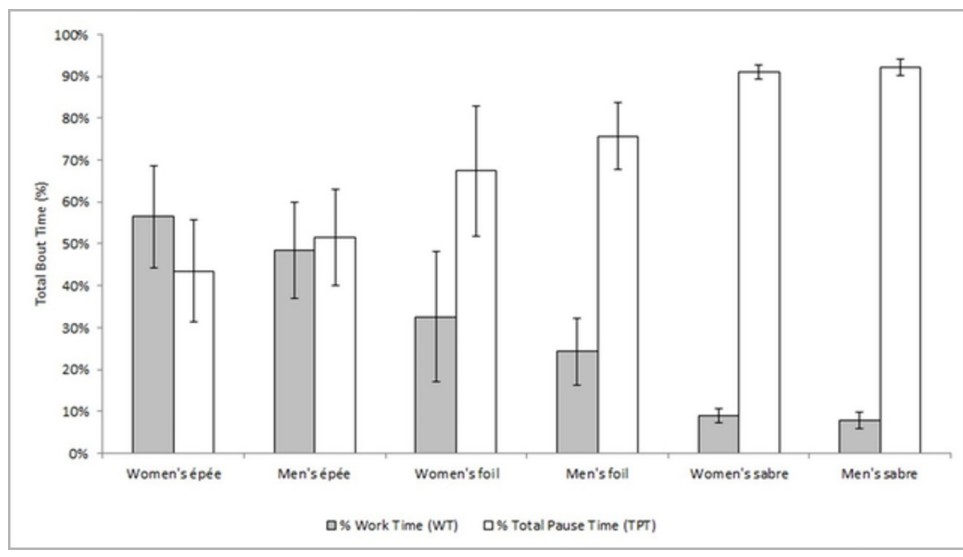

**Fig 1. Relationship between % Work Time (WT·[WT+TPT]$^{-1}$) and % Total Pause Time (TPT·[WT+TPT]$^{-1}$) in bouts of the final phase of 2014 World Fencing Championships.**

**Table 2. Temporal structure of the final phase of 2014 World Épée Championship according to the bout periods.**

| | 1PER | | | 2PER | | | 3PER | | | ET_A | | | ET_B | | | |
|---|---|---|---|---|---|---|---|---|---|---|---|---|---|---|---|---|
| **Men's épée** | | | | | | | | | | | | | | | | χ2(2) |
| **Bouts (n)** | | 15 | | | 15 | | | 14 | | | 3 | | | 1 | | |
| **TBT (s)** | 363.6 | ± | 135.2 | 372.0 | ± | 134.2 | 284.7 | ± | 192.1 | 21.7 | ± | 8.4 | 112.3 | | | 12.196 αγ$ |
| **WT (s)** | 162.0 | ± | 43.3 | 163.1 | ± | 38.2 | 119.7 | ± | 61.2 | 14.4 | ± | 4.4 | 58.4 | | | 12.981 αγ$ |
| **TPT (s)** | 201.7 | ± | 100.4 | 208.9 | ± | 105.6 | 165.0 | ± | 140.6 | 22.1 | | | 53.9 | | | 10.787 αγ$ |
| **PRT (s)** | 140.4 | ± | 74.2 | 145.6 | ± | 84.5 | 156.6 | ± | 141.6 | 22.1 | | | 53.9 | | | |
| **PTBP (s)** | 73.8 | ± | 22.3 | 78.3 | ± | 20.0 | 29.5 | ± | 10.9 | | | | | | | 29.422 &#γ |
| *Halte* **(n)** | 8.5 | ± | 4.2 | 7.8 | ± | 2.7 | 8.6 | ± | 4.8 | 1.3 | ± | 0.6 | 4.0 | | | |
| **AAT (s)** | 25.0 | ± | 14.0 | 23.8 | ± | 10.3 | 15.1 | ± | 6.6 | 12.8 | ± | 7.1 | 14.6 | | | 12.733 &#γ |
| **AHT (s)** | 16.9 | ± | 4.7 | 19.5 | ± | 9.5 | 19.0 | ± | 9.5 | 22.1 | | | 18.0 | | | |
| **W/R** | | 1: 0.7 | | | 1: 0.8 | | | 1: 1.3 | | | 1: 1.7 | | | 1: 1.2 | | |
| **Women's épée** | | | | | | | | | | | | | | | | χ2(2) |
| **Bouts (n)** | | 15 | | | 15 | | | 14 | | | 2 | | | 2 | | |
| **TBT (s)** | 283.9 | ± | 109.3 | 313.4 | ± | 105.5 | 230.7 | ± | 112.5 | 70.0 | ± | 52.3 | 143.0 | ± | 22.9 | 9.904 |
| **WT (s)** | 150.8 | ± | 49.0 | 155.7 | ± | 40.6 | 122.6 | ± | 54.4 | 33.6 | ± | 17.1 | 62.9 | ± | 1.2 | 10.003 αγ$ |
| **TPT (s)** | 133.1 | ± | 63.6 | 157.7 | ± | 73.1 | 108.1 | ± | 65.0 | 36.4 | ± | 35.2 | 80.1 | ± | 21.7 | n.s. |
| **PRT (s)** | 73.9 | ± | 29.7 | 99.2 | ± | 40.7 | 104.4 | ± | 60.8 | 36.4 | ± | 35.2 | 80.1 | ± | 21.7 | n.s. |
| **PTBP (s)** | 69.1 | ± | 31.4 | 69.7 | ± | 31.2 | 39.0 | ± | 11.3 | | | | | | | 26.196 &#αβγδ |
| *Halte* **(n)** | 6.27 | ± | 3.4 | 8.7 | ± | 3.8 | 8.1 | ± | 4.2 | 3.0 | ± | 1.4 | 9.5 | ± | 0.7 | n.s. |
| **AAT (s)** | 30.4 | ± | 15.0 | 21.1 | ± | 10.4 | 17.5 | ± | 7.9 | 11.1 | ± | 0.5 | 6.6 | ± | 0.4 | 17.060 *&αβγδ$ |
| **AHT (s)** | 14.1 | ± | 7.3 | 12.2 | ± | 3.4 | 13.1 | ± | 3.2 | 16.0 | ± | 6.3 | 9.3 | ± | 1.8 | n.s. |
| **W/R** | | 1: 0.5 | | | 1: 0.6 | | | 1: 0.7 | | | 1: 1.4 | | | 1: 1.4 | | |

Note. Kruskal-Wallis test χ2(2) p < .05: 1PER > 2PER

*; 2PER > 3PER

#; 1PER > 3PER &; 1PER > ET_A α; 1PER > ET_B β; 2PER > ET_A

γ; 2PER > ET_B

δ; 3PER < ET_A $.

the differences (p < .05) in the average work time (AAT) between *Allez* and *Halte* is reduced as the bout progresses such that the AAT is less in the second period than the first and less again in the third period (Table 2).

In Table 2 it can be seen how the temporal records in men's épée are similar to those of women's épée, however, men show shorter rest times (TPT & AHT) than women (p < .05), which implies that their work times (WT) represent approximately 45% of the total fight time (TBT). Men's épée shows the same tendency for fight times (AAT) to decrease as the periods advance, but not with the same level of statistical significance.

Table 3 presents the results of foil where we can see that both women and men have lower (p < .05) fight action times (WT & AAT) than épée and higher than sabre representing values close to 25% of work time in the assault (WT) in relation to the total time of the fights (TBT). In women's foil, a decrease in work and rest times (WT & TPT) between the first and the second period is observed (p < .05). In men's foil matches, similar records can be seen to women's foil in relation to the different modalities of analysis, with no significant differences between fencers of different genders in foil. This is contrary in épée in which we observed that 93% of the fights reach the 3rd period, whereas in foil, only 17% of the observed fights reached the third period.

**Table 3. Temporal structure of the final phase of 2014 World Foil& Sabre Championship according to the bout periods.**

| | 1PER | | | 2PER | | | 3PER | | | χ2(2) | 1PER | | | 2PER | | | 3PER | | | χ2(2) |
|---|---|---|---|---|---|---|---|---|---|---|---|---|---|---|---|---|---|---|---|---|
| | Women's foil | | | | | | | | | | Men's foil | | | | | | | | | |
| Bouts (n) | 11 | | | 10 | | | 2 | | | | 12 | | | 7 | | | 2 | | | |
| TBT (s) | 630.8 | ± | 230.0 | 303.5 | ± | 263.7 | 470.3 | ± | 254.5 | 6.704 * | 710.5 | ± | 149.6 | 378.9 | ± | 272.6 | 275.3 | ± | 257.0 | 9.668 *& |
| WT (s) | 176.3 | ± | 20.1 | 83.5 | ± | 67.9 | 155.9 | ± | 45.6 | 9.622 * | 163.1 | ± | 29.8 | 94.8 | ± | 70.6 | 92.8 | ± | 97.9 | 6.939 * |
| TPT (s) | 454.5 | ± | 65.6 | 220.0 | ± | 212.1 | 314.3 | ± | 209.0 | 9.622 * | 547.3 | ± | 161.8 | 284.1 | ± | 220.0 | 182.5 | ± | 159.1 | 8.742 *& |
| PRT (s) | 376.6 | ± | 31.7 | 200.2 | ± | 212.4 | 314.3 | ± | 209.0 | | 496.3 | ± | 185.4 | 259.0 | ± | 208.9 | 182.5 | ± | 159.1 | 9.668 *& |
| PTBP (s) | 85.7 | ± | 28.8 | 99.0 | ± | 9.3 | | | | 8.546 * | 87.6 | ± | 10.9 | 88.1 | ± | 3.4 | | | | |
| *Halte* (n) | 28.1 | ± | 11.0 | 12.8 | ± | 10.7 | 21.0 | ± | 15.6 | 6.653 * | 34.4 | ± | 9.5 | 14.4 | ± | 9.0 | 12.0 | ± | 7.1 | 11.226 *& |
| AAT (s) | 9.3 | ± | 10.0 | 9.2 | ± | 12.5 | 9.1 | ± | 4.6 | | 5.4 | ± | 2.6 | 6.5 | ± | 3.9 | 6.5 | ± | 4.4 | |
| AHT (s) | 13.7 | ± | 4.2 | 16.9 | ± | 5.9 | 16.7 | ± | 2.5 | | 14.9 | ± | 3.2 | 18.1 | ± | 6.6 | 15.1 | ± | 4.8 | |
| W/R | 1: 1.5 | | | 1: 1.8 | | | 1: 1.8 | | | | 1: 2.8 | | | 1: 2.8 | | | 1: 2.3 | | | |
| | Women's Sabre | | | | | | | | | χ2(2) | Men's Sabre | | | | | | | | | χ2(2) |
| Bouts (n) | 15 | | | 15 | | | | | | | 15 | | | 15 | | | | | | |
| TBT (s) | 360.7 | ± | 89.3 | 355.0 | ± | 123.4 | | | | | 413.5 | ± | 109.9 | 413.5 | ± | 233.9 | | | | |
| WT (s) | 32.8 | ± | 10.8 | 37.5 | ± | 12.4 | | | | | 31.9 | ± | 8.7 | 31.4 | ± | 13.0 | | | | |
| TPT (s) | 327.9 | ± | 80.9 | 317.6 | ± | 113.4 | | | | | 387.4 | ± | 68.6 | 382.1 | ± | 224.7 | | | | |
| PRT (s) | 231.1 | ± | 73.9 | 317.6 | ± | 113.4 | | | | 4.047π | 291.2 | ± | 88.5 | 382.1 | ± | 224.7 | | | | |
| PTBP (s) | 96.8 | ± | 15.6 | | | | | | | | 96.2 | ± | 39.6 | | | | | | | |
| *Halte* (n) | 17.9 | ± | 5.0 | 19.5 | ± | 5.1 | | | | | 21.1 | ± | 6.2 | 21.2 | ± | 11.2 | | | | |
| AAT (s) | 1.8 | ± | 0.4 | 1.9 | ± | 0.4 | | | | | 1.6 | ± | 0.5 | 1.7 | ± | 0.8 | | | | |
| AHT (s) | 13.8 | ± | 2.9 | 16.9 | ± | 3.5 | | | | 6.297π | 14.5 | ± | 2.2 | 19.0 | ± | 5.1 | | | | 8.073π |
| W/R | 1: 7.5 | | | 1: 8.8 | | | | | | | 1: 9.2 | | | 1: 11.3 | | | | | | |

Note. Kruskal-Wallis test χ2(2) p < .05: 1PER > 2 PER

*; 1PER < 2PER π; 1PER > 3PER &

Sabre bouts present working time percentages (WT) of less than 10% of the total bout time (TBT), with time records for both WT and AAT being much lower in sabre than in foil and épée. In relation to the mean activity times between allez and halte (AAT), in men's and women's sabre, unlike épée, the mean values of the 2nd period are similar to those of the first period, while in foil and épée they are not. Differences can be seen in the average rest times in the rounds (AHT), with sabre having greater fight stopping times in the second period in relation to the first (Table 3).

## Discussion

This is the first study to analyse all six modalities of fencing in the same competition and to compare the temporary structure in international elite fencers. The first studies were carried out at the 1981 World Championship [1, 3]. At this time the direct elimination rounds were ten hits and 10 minutes. In the 1990s, direct elimination began to be played in two or three rounds with 5 hits, until reaching the current system in which the winner must complete 15 hits or have the highest number of hits at the end of 9 minutes, and the additional minute, if necessary [16]. We must consider the existing differences in global times (TBT, WT, TPT...) as a product of different rules on the duration of the bouts over several decades. However, the average values (AAT and AHT) as well as the work-to-rest ratios are not necessarily affected by the change in fight time or the number of hits. A summary table of this data can be found in Table 4.

**Table 4. Comparative of temporal structure in official competitions.**

| Weapon | Data | | TBT (s) | | | WT (s) | | | TPT (s) | | | AAT (s) | | | AHT (s) | | | W/R |
|---|---|---|---|---|---|---|---|---|---|---|---|---|---|---|---|---|---|---|
| WE | World Championship 2014 | * | 841.0 | ± | 214.3 | 433.8 | ± | 103.8 | 407.2 | ± | 121.2 | 17.8 | ± | 2.8 | 12.4 | ± | 2.0 | 1: 0.7 |
| | Pittaluga & Roi (1999) | α | 623.0 | ± | 151.0 | 366.0 | ± | 109.0 | 263.0 | ± | 71.0 | 16.5 | ± | 4.2 | 7.9 | ± | 2.7 | 1: 0.5 |
| ME | World Championship 2014 | * | 1016.1 | ± | 284.6 | 444.3 | ± | 88.0 | 571.9 | ± | 215.9 | 18.1 | ± | 3.5 | 18.6 | ± | 6.2 | 1: 1.0 |
| | Pittaluga & Roi (1999) | α | 728.0 | ± | 247.0 | 302.0 | ± | 86.0 | 425.0 | ± | 162.0 | 12.7 | ± | 7.6 | 18.2 | ± | 12.3 | 1: 1.4 |
| | World Championship 1981 | β | | 570.0 | | | 410.0 | | | 160.0 | | | 18.5 | | | 11.5 | | 1: 0.6 |
| WF | World Championship 2014 | * | 992.3 | ± | 316.9 | 280.6 | ± | 112.5 | 711.6 | ± | 253.0 | 6.5 | ± | 2.6 | 14.5 | ± | 3.6 | 1: 2.2 |
| | World Championship 1981 | β | | 615.0 | | | 310.0 | | | 305.0 | | | 8.7 | | | 9.7 | | 1: 1.1 |
| MF | World Championship 2014 | * | 977.4 | ± | 322.0 | 233.9 | ± | 123.8 | 743.5 | ± | 236.9 | 5.1 | ± | 2.3 | 15.6 | ± | 3.8 | 1: 3.0 |
| | Pittaluga & Roi (1999) | α | 997.0 | ± | 227.0 | 253.0 | ± | 82.0 | 745.0 | ± | 145.0 | 5.2 | ± | 3.5 | 15.6 | ± | 12.8 | 1: 3.0 |
| | World Championship 1981 | β | | 660.0 | | | 338.0 | | | 322.0 | | | 8.6 | | | 11.0 | | 1: 1.3 |
| WS | World Championship 2014 | * | 715.7 | ± | 156.9 | 70.3 | ± | 16.6 | 645.4 | ± | 143.2 | 1.9 | ± | 0.3 | 15.5 | ± | 2.5 | 1: 8.2 |
| | World Cup 2009–10 | $ | 417.9 | ± | 99.5 | 71.6 | ± | 21.8 | 346.3 | ± | 93.7 | 2.9 | ± | 0.9 | 14.5 | ± | 3.2 | 1: 5.1 |
| MS | World Championship 2014 | * | 832.8 | ± | 286.0 | 63.3 | ± | 14.6 | 769.5 | ± | 276.8 | 1.6 | ± | 0.5 | 16.6 | ± | 3.4 | 1: 10.4 |
| | World Cup 2009–10 | $ | 516.2 | ± | 81.6 | 70.7 | ± | 17.2 | 445.5 | ± | 72.8 | 2.5 | ± | 0.6 | 16.5 | ± | 2.7 | 1: 6.5 |
| | World Championship 1981 | β | | 525.0 | | | 230.0 | | | 295.0 | | | 5.2 | | | 9.4 | | 1: 1.8 |

Note.

* Our data; α Data of Pittaluga & Roi [5] in Aquilli [7]

$ Aquilli et al. [7]; β Marini [3] and Lavoie et al. [1].

Marini [3] analysed the 1981 World Fencing Championships, concluding that the WT represented the 71.9% of the TBT for ME, values very different from those found by Pittaluga and Roi [5], 41.5%, and us 43.7%. We need to consider that these studies were carried out 30 years before, when the bouts were only to 10 hits and with a duration of 10 minutes. For WE bouts, Pittaluga and Roi [5] observed the WT to be 58.7% of the TBT, and we found it to be 51.6%. We do not have any previous records of women's épée which was incorporated for the first time into a world championship in 1989 [16].

The results of sabre were the other extreme, with 9.8% in WS and 7.6% in MS, something far from the results of Aquili et al. [7], which were 17.1% for WS and 13.7% for MS, and even more for those of Marini [3], 43.8% for MS. Sabre rules have changed significantly over the years especially as there was no electrical signalling system in 1984 for sabre and fencers were allowed to cross their legs (fleche) during the fight. We do not have any previous records since women's sabre was incorporated for the first time into a world championship in 1999 [16].

Foil is placed in an intermediate term, representing the WT 28.3% of the TBT in WF and 23.9% in MF. These values are very similar to those obtained for MF by Pittaluga and Roi [5], which was 25.4%, although very different from 50.4% and 51.2% obtained by Marini [3], and 63% and 54% recorded by Waterloh et al. [4] for WF and MF respectively. In the case of foil, we can see how the work-to-rest ratio is almost doubled in favour of rest between the two world championships. A probable cause could also be a regulatory change, in particular the required time was modified in foil for the achievement of a double hit by the two fencers, going from 800 ms in 1981 to 300 ms in 2014.

Perhaps the most interesting finding of this study is related to the Allez mean time (AAT). The values range between 1.6 s (±0.5) for MS and 18.1 s (±3.5) for ME, showing significant differences ($p < .05$) in both with the rest of the disciplines of fencing, but without gender differences in the same weapon.

The activity values (AAT) are very different in the three weapons, being less than 2 seconds in sabre, close to 6 seconds in foil and 18 seconds in épée. Our results agree with Aquili et al.

[7] on some of the differences between men's sabre and women's sabre. Men have higher values than women in TBT and TPT, while women have higher values than men in WT and AAT. The W/R is also higher in men in both studies. However, Aquili et al. [7] found significant differences between male and female sabre in TBT, TPT and W/R, while in our work no significant differences were found between men and women's sabre in any of the time records. In sabre, both opponents want to have the initiative, which is why sabre has become a weapon with a lot of physical power, with immediate and very explosive actions motivated by the regulations and the new blocking times of the apparatus. In the appearance of the electric sabre in the 90s, the detection of the double hit occurred with a difference between hits of 400 ms. In our study, the FIE rules reduced these values to 130 ms.

The description of the W/R in women's foil of 1:1.1 made by Wylde et al. [9] does not exactly respond to the periods of work and rest that we have described in this work. While we analysed the pause time between the Halt and Allez/Play referee's calls, these authors performed an analysis comparing the movements according to intensity and considered low-intensity actions not only during the pauses between hits. One of the factors that affects the work and rest times is non-combativity–when a minute passes without a hit scored. Decades ago, this concept did not exist. This issue mainly occurs in épée. Specifically, in ME we observed four periods of 3 minutes with non-combativity, and six in WE.

The temporal parameter that has the most similarity between the three weapons is the AHT, which reached its minimum value in the first period for the six modalities. Iglesias et al. [10] gave the following explanation for this circumstance, at the end of the bouts the fencers are under increasing environmental pressure (influence of space, time, scoreboard...), which can lead them to take a longer rest each time the bout is stopped, in order to have more time for tactical thinking.

The fight work-to-rest ratios recorded in the present study were 1:0.9, 1:2.6 and 1:9.2 for épée, foil and sabre respectively. These differences between weapons could suggest different energy systems are being utilised during competition. Oates et al. [17] has highlighted that blood lactate concentrations across all weapons are below the onset of blood lactate accumulation (4mmol.L$^{-1}$) with foil ~ 2.4 mmol·L$^{-1}$ [18] and épée ~2.7 mmol·L$^{-1}$ [19], suggesting a relatively low reliance on the lactic acid system. The W/R of sabre is the lowest at 1:0.9 which suggests there is a greater reliance on the alactic energy system compared to épée and foil. However, several authors have a general agreement that all weapons rely on the alactic energy system to provide explosive movements such as the lunge [11, 19, 20]. There has been one study by Yang et al. [21] which has specifically reported the energetics of épée fencing and found that 80–90% of a fight utilises the aerobic energy system and increases as a fight progresses. Further research is required on the energetics of fencing to fully understand the differences between weapons.

## Practical applications

The differences between weapons in relation to the W/R may be a factor in considering the conditioning programme for fencers. Turner et al. [11, 12] determined that the technical and tactical differences between each weapon may in part explain some of the variance in the temporal parameters, but these differences do not necessarily involve specific strength and conditioning training. Whereas some authors [1, 20] contradicted this by indicating that the fencing coaches should focus on replicating the demands of fencing competition to train their fencers. For this reason, knowing the W/R and other specific temporal parameters of each weapon ensures the best preparation of the fencers and helps to simulate the demands of a competitive bout.

Wylde et al. [9] suggests, in foil, conditioning programs should focus mainly in alactic anaerobic activities. It also proposes to develop aerobic competition to improve recovery between hits and bouts. These assessments agree with those of a previous study of women's épée fencing [20] but contradicts with Turner et al. [13] who suggests similar conditioning for all weapons. The work-to-rest values from the current study are significantly different with sabre being 1 .7 ± 0.4 s and épée being 17.9 ± 3.1 s which suggests conditioning should be different between weapons to prepare them for competition. However, the current results demonstrate that there is no need to differentiate between the physical training of women and men of the same weapon.

## Conclusion

We conclude that there are significant differences in the work and rest times between the three fencing weapons. Likewise, there are no differences between fencers of the same weapon according to gender, except for lower rest values in women's épée compared to men's. There are significant differences between the working times in the three fencing weapons, highlighting the great differences in the mean values observed, which are 17.9 ± 3.1 s for épée, 5.8 ± 2.5 s for foil and 1 .7 ± 0.4 s for sabre. Finally, the work-to-rest ratios recorded in the final phases of the fencing world championship was 1:0.9 in épée, 1:2.6 in foil and 1:9.2 in sabre.

## Acknowledgments

The authors would like to thank Agustí Gasset and Juanjo Michavila for their assistance in the initial phases of this study.

## Author Contributions

**Conceptualization:** Rafael Tarragó, Xavier Iglesias.

**Data curation:** Rafael Tarragó, Xavier Iglesias.

**Formal analysis:** Rafael Tarragó, Lindsay Bottoms, Xavier Iglesias.

**Methodology:** Rafael Tarragó, Lindsay Bottoms, Xavier Iglesias.

**Project administration:** Rafael Tarragó, Xavier Iglesias.

**Software:** Rafael Tarragó, Xavier Iglesias.

**Writing – original draft:** Rafael Tarragó, Lindsay Bottoms, Xavier Iglesias.

**Writing – review & editing:** Rafael Tarragó, Lindsay Bottoms, Xavier Iglesias.

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
