## [Decision Letter · Decision Letter 0]

27 Mar 2023

PONE-D-23-04357Work-to-rest Ratios in Elite FencingPLOS ONE

Dear Dr. Bottoms,

Thank you for submitting your manuscript to PLOS ONE. After careful consideration, we feel that it has merit but does not fully meet PLOS ONE’s publication criteria as it currently stands. Therefore, we invite you to submit a revised version of the manuscript that addresses the points raised during the review process.

We look forward to receiving your revised manuscript.

Kind regards,

Monika Błaszczyszyn

Academic Editor

PLOS ONE

Journal Requirements:

Reviewers' comments:

Reviewer's Responses to Questions

**Comments to the Author**

1. Is the manuscript technically sound, and do the data support the conclusions?

Reviewer #1: Yes

Reviewer #2: Yes

2. Has the statistical analysis been performed appropriately and rigorously? 

Reviewer #1: Yes

Reviewer #2: Yes

3. Have the authors made all data underlying the findings in their manuscript fully available?

Reviewer #1: Yes

Reviewer #2: Yes

4. Is the manuscript presented in an intelligible fashion and written in standard English?

Reviewer #1: Yes

Reviewer #2: Yes

5. Review Comments to the Author

Reviewer #1: The subject of the study is very interesting. I am very pleased that these kinds of research investigations are also being created. In my opinion, thanks to such studies we can better understand the load of athletes in fencing. These findings can be applied within the training environment and performance enhancement. I also believe that the analysis of temporal relationships within weapons will help other potential researchers to follow in detail, for example, the physiological aspects of the load during matches or training. Adjusting training according to the data obtained can ensure performance growth. The authors rightly point out that current knowledge in this issue is based on research information that is "relatively" old. It is necessary to mention that today's fencing is different from the fencing that was practiced 30 years ago. It can be argued that fencing has definitely gotten faster.

The title of the manuscript is too specific, focused on one part of the solved problem. Wouldn't it be better to introduce Analysis of the Time Interactions in Elite Fencing? In my opinion, words should not be repeated in the article title and keywords. The abstract is apt. The only thing I would modify: it is not clear what the average working time and work-to-rest ratios are. The results in Abstract are detailed, but there is no explanation of the time intervals. Maybe it should generalized in this part. Perhaps at the end of the abstract I would emphasize where the results can be applied.

In the Introduction, the authors rely on relevant data and it is possible to deduce exactly what goal they are pursuing.

Methodology:

I would point out here that this type of research can be done and facilitated through machine learning/artificial intelligence. A large number of records can cause errors ("human mistakes"). The beginning and end moments of the analyzed time periods could have been better defined. The number of matches analyzed is admirable.

The results are presented clearly.

It is important to realize that the length of the matches, the intensity of the matches, etc. are affected (in connection with the presented study) by a number of other factors: the quality of the opponent, the surrounding environment, the referee, the importance of the match (final vs. 1st elimination match), the moment in time within day etc.

In further studies, it would be interesting to also determine the number of individual attacks or defensive maneuvers. These data could be valuable compared with the results of previous studies (20, 30 years ago). Thanks to such results, coaches and fencers could transform training plans and resources.

Reviewer #2: Dear authors,

This is generally a well written manuscript. It is an interesting study that covers a very specific and important topic for the scientific and sport community, into a previously under researched area providing new data and findings. Overall, this is a paper that could eventually make a useful contribution to the literature

Although simple in methodology, it advances our knowledge on whether different factors such as gender and weapon may affect time analysis of elite fencers

The introduction is to the point. Although there is limited information in this topic, the review of the literature is adequate giving sufficient, extensive and detailed attention to the time motion fencing analysis, including almost all variables seem to be important in fencing competition

What is included in the methods is generally clear and sufficiently detailed.

The results are both presented statistically and substantively meaningful. Tables Figures and their relative legends are quite informative.

In the discussion section authors highlighted the importance of this paper and answered all the questions set out and did a good job of synthesizing the literature.

The writing style is clear, concise and correct.

A light but balanced critic in some points of the article will help the reader to associate the existing literature and the necessity of the future studies. Please, improve in the discussion section the relationship between time analysis and fencing energetics. Can we propose different energetic mechanisms for each weapon? No more than a paragraph, please.

What impressed me in this review are the conclusions in which the authors convince me that they are deep experts of the fencing science. The future researchers have to take into consideration that the necessary physical and functional demands of fencing are multifactorial and cannot be evaluated separately in the laboratory, but rather in association with the opponent’s behavior.

6. PLOS authors have the option to publish the peer review history of their article (what does this mean?). If published, this will include your full peer review and any attached files.

Reviewer #1: **Yes: **Štefan Balkó

Reviewer #2: No

---

## [Author Response · Author response to Decision Letter 0]

12 Apr 2023

Thank you for reviewing our paper. We have gone through the feedback and tried to make the amendments suggested. Hopefully this has improved the manuscript. 

Reviewer1: Reviewer #1: The subject of the study is very interesting. I am very pleased that these kinds of research investigations are also being created. In my opinion, thanks to such studies we can better understand the load of athletes in fencing. These findings can be applied within the training environment and performance enhancement. I also believe that the analysis of temporal relationships within weapons will help other potential researchers to follow in detail, for example, the physiological aspects of the load during matches or training. Adjusting training according to the data obtained can ensure performance growth. The authors rightly point out that current knowledge in this issue is based on research information that is "relatively" old. It is necessary to mention that today's fencing is different from the fencing that was practiced 30 years ago. It can be argued that fencing has definitely gotten faster.

Response: Thank you for the positive comments about our paper. 

Reviewer 1: The title of the manuscript is too specific, focused on one part of the solved problem. Wouldn't it be better to introduce Analysis of the Time Interactions in Elite Fencing? 

Response: Thank you for the suggestion, we have now amended the title to ‘Temporal demands of elite fencing’ to reflect this. 

Reviewer 1: In my opinion, words should not be repeated in the article title and keywords. 

Response: We have now changed the keywords to be different to those in the title. 

Reviewer 1: The abstract is apt. The only thing I would modify: it is not clear what the average working time and work-to-rest ratios are. 

Response: We have provided a definition of work time which hopefully improves the clarity. 

Reviewer 1: The results in Abstract are detailed, but there is no explanation of the time intervals. Maybe it should generalized in this part. Perhaps at the end of the abstract I would emphasize where the results can be applied.

Response: Thank you for the comment. Hopefully the explanation of work time helps clarify this. As much as we agree with you that it would be good to emphasise how the results can be applied, we are constrained by the abstract word count. 

Reviewer 2: Please, improve in the discussion section the relationship between time analysis and fencing energetics. Can we propose different energetic mechanisms for each weapon? No more than a paragraph, please.

Response: We have added an additional paragraph starting line 273 on page 16 which hopefully expands on this.

---

## [Editor Report · Decision Letter 1]

14 Apr 2023

Temporal Demands of Elite Fencing

PONE-D-23-04357R1

Dear Dr. Bottoms,

We’re pleased to inform you that your manuscript has been judged scientifically suitable for publication and will be formally accepted for publication once it meets all outstanding technical requirements.

Kind regards,

Monika Błaszczyszyn

Academic Editor

PLOS ONE

---

## [Editor Report · Acceptance letter]

22 May 2023

PONE-D-23-04357R1 

Temporal Demands of Elite Fencing 

Dear Dr. Bottoms:

I'm pleased to inform you that your manuscript has been deemed suitable for publication in PLOS ONE. Congratulations! Your manuscript is now with our production department. 

Kind regards, 

on behalf of

Dr. Monika Błaszczyszyn 

Academic Editor

PLOS ONE